# Eight Weeks of High-Intensity Interval Training Using Elevation Mask May Improve Cardiorespiratory Fitness, Pulmonary Functions, and Hematological Variables in University Athletes

**DOI:** 10.3390/ijerph20043533

**Published:** 2023-02-17

**Authors:** Nasser Abouzeid, Mahmoud ELnaggar, Haytham FathAllah, Mostafa Amira

**Affiliations:** 1Department of Physical Education, College of Education, King Faisal University, Al-Ahsa 31982, Saudi Arabia; 2Department of Games Sports and Racquet Games Training, College of Physical Education for Boys, Helwan University, Cairo 11795, Egypt; 3Department of Sports Health Sciences, College of Physical Education, Helwan University, Cairo 11795, Egypt; 4Sports Training Department, Faculty of Physical Education, Zagazig University, Zagazig 44519, Egypt

**Keywords:** elevation training masks, HIIT, endurance training, cardiorespiratory fitness, hematology

## Abstract

Background: In the last two decades, high-altitude training (HAT) and elevation training masks (ETMs) have been widely used among athletes to enhance physical performance. However, few studies have examined the effect of wearing ETMs on physiological and hematological parameters in different sports. Aims: The present study aimed to investigate the impact of ETM use in athletes on several hematological and physiological indicators among cyclists, runners, and swimmers. Methods: The impact of wearing an ETM on lung function (LF), aerobic capacity (AC), and hematological levels in male university-level athletes (cyclists, runners, and swimmers) was investigated using an experimental approach. The participants (N = 44) were divided into (i) an experimental group wearing ETMs (n = 22; aged 21.24 ± 0.14 years old) and (ii) a control group not wearing ETMs (n = 22; aged 21.35 ± 0.19 years old). Both groups underwent 8 weeks of high-intensity cycle ergometer interval training. Pre- and post-training tests included the above-mentioned physiological and hematological parameters. Results: Except for FEV₁, FEV₁/FVC, VT1, and MHR in the control group and FEV₁/FVC and HRM in the experimental group, all variables were significantly improved after the 8-week cycle ergometer HIIT program. Significant benefits in favor of the experimental group were noted in terms of changes in FVC, FEV₁, VO₂max, VT1, PO to VT, VT2, and PO to VT2. Conclusions: The eight-week ETM-assisted HIIT program improved cardiorespiratory fitness and hematological variables in all participants. Future research would be useful to further investigate the physiological changes resulting from ETM-assisted HIIT programs.

## 1. Introduction

Over the last twenty years, athletes have employed a range of elevation training masks (ETMs) (designed to simulate altitude training) to achieve improvements in aerobic fitness and physical performance [1,2]. ETMs restrict the flow of air on inhalation and exhalation during exercise via flux valves that can be adjusted to limit the volume of air [2]. A common misconception among athletes is that wearing an ETM simulates exercise at altitude, creating a hypobaric effect (decreasing partial oxygen pressure); there is robust evidence that training with an ETM does not lead to hypoxia as measured by improved hematological parameters [1,3,4]. Training at high altitudes is difficult and expensive and so is not often used by amateur athletes [5,6]. Compared to altitude training, ETMs are more cost-effective and easily accessible for athletes as they produce comparable effects to altitude training in terms of enhancing athletic performance [7].

ETMs feature adjustable valves that cover the mouth and nose to restrict breathing [3,7]. Previous studies have investigated the impact of wearing ETMs during training programs in amateur runners and cyclists participating in HIIT and Reserve Officers’ Training Corps (ROTC) cadets undergoing physical training (PT) [7,8,9,10,11,12]. The results of these studies were mixed in terms of the capacity of ETMs to enhance VO2peak compared to the non-ETM-wearing control groups [1,2,3,4]. Two studies found that ETM use enhanced VO2peak post-training, yet the difference between the two groups was not significant [7,8]. This may have been due to disparities in the measurement of VO2peak among the four studies and the variations between the intensity of the HIIT and the ROTC PT sessions. Another experimental study found that both ETM use and no ETM use in HITT cycling sessions significantly enhanced PPO in both groups [2]. ETM use improved aerobic and anaerobic thresholds and PO (measured pre- and post-training); this was not the case in non-ETM users; ETM use appeared to cause significantly lower blood oxygen saturation (SpO2) levels in wearers compared to the control group (94% vs. 96%, respectively) [7]. These results were confirmed by another study that modified the restrictive breathing mode of ETMs with a control group (not wearing ETMs) to replicate exercising (jogging) at 2743 and 4572 m above sea level at 60% of VO2max. In the experimental group, SpO2 levels ranged from 94 to 91% and 89% in the control group at 2743 m and 4572 m, respectively [3]. Ideally, professional athletes should train in high-altitude hypoxic conditions throughout all stages of preparation. Training for short track-based events does not tend to involve hypoxic training at altitude due to the nature of such events and the lack of training facilities at high-altitude locations (>1800 m) above sea level. Enhancing the efficacy of such training requires constant research to develop novel evidence-based methods, including using diagnostic approaches to gather data on functional efficiency indicators. This is particularly relevant for elite athletes, as any slight advantage over their competitors conferred by novel training methods can be critical for success. In light of the above, the present study aimed to investigate the impact of eight weeks of HIIT using ETMs on physiological and hematological variables in university athletes. This experimental study is the first phase of our project, the outcomes of which will be compared with the outcomes of the second phase, in which the same participants will be tested in a simulated-altitude hypoxic environmental chamber.

## 2. Materials and Methods

### 2.1. Study Design

This experimental study examined the effect of ETM wearing on aerobic capacity (AC), lung function (LF), and hematological indicators among male university athletes (runners: n = 16; swimmers: n = 14; cyclists n = 14). The participants were divided into two comparable groups: Group 1 (experimental group) ETM wearers (N = 22; runners: n = 8; swimmers: n = 7; cyclists: n = 7; aged 21.24 ± 0.14 years old mean ± SD); Group 2 (control group) non-ETM wearers (N = 22; runners: n = 8; swimmers: n = 7; cyclists: n = 7; aged 21.35 ± 0.19 years old mean ± SD).

### 2.2. Participants

Forty-four male athletes were recruited voluntarily from the King Faisal University campus in Al-Ahsa, Saudi Arabia. Based on work by Lundby et al. [7], the required sample size (N = 44) was calculated using the G-Power program (version 3.1.9.7); 22 participants were required to detect a moderate effect size (0.5) in VO2peak with a power of 80% and a probability value greater than 0.05 (*p* > 0.05). A total of 25 participants were recruited to accommodate an expected dropout rate of about 10%. Only 22 participants from each group completed the entire training program and all measurement stages. All participants were informed about the experimental procedures. The researcher explained the purpose, procedures, and potential risks of participating in the study; all participants signed a written consent form. The study was approved by the Research Ethical Committee and performed in accordance with the Declaration of Helsinki at the College of Education at King Faisal University.

### 2.3. Anthropometry

Participants’ height (to the nearest 0.1 cm) was measured using a height rod (Deteco Electronic, Model: DHRWM, USA), and weight (to the nearest 0.1 kg) was measured using a Seca weight scale (Seca-869, Germany). Body mass index (BMI) was then calculated (body mass in kg/height in meters squared (kg.m^−2^)).

### 2.4. Blood Parameters

Hematocrit (Hct) levels were measured in all participants via the use of a capillary tube and a microhematocrit centrifuge (International Equipment Co., Needham Heights, MA, USA); hemoglobin concentrations (Hb) were measured via the use of a hemoglobin reagent set and hemoglobin standard (Pointe Scientific Inc., Canton, MI, USA) and spectrophotometric analysis (Spectronic 20D+, Thermo Spectronic, Rochester, NY, USA).

### 2.5. Procedures

First, a pilot trial was conducted (n = 3) to determine an appropriate training protocol and familiarize the participants with using an ETM during exercise. Data were gathered from Monark 828E Ergomedic cycle ergometers (Monark Exercise AB, Vansbro, Sweden) at King Faisal University’s medical center. At first, all participants (control and experimental groups) underwent maximal cycle ergometer testing to determine their respective VO2peak, VT1, VT2, maximal heart rate (HRpeak), and PPO. VO2peak testing was carried out using an electronically braked cycle ergometer (Lode B.V., Groningen, The Netherlands). The workload was set at 25 W for 3 min; PO was then increased incrementally by 25 W per minute until fatigue set in. Respiratory gas exchange was measured using an open-circuit, mixing-chamber-type spirometry system (AEI Technologies, Naperville, IL, USA).

Heart rate was measured at 1 min intervals using Polar watches (Polar Vantage XL, Polar Instruments, Port Washington, NY, USA). Participants rated their perceived exertion (RPE) at 1 min intervals using the modified Borg CR-10 scale [13]. Ventilatory threshold-1 (VT1) and Ventilatory threshold-2 (VT2) were investigated by integrating the ventilatory equivalent method and the V-slope method [14]. Pulmonary function was measured in both groups. Forced vital capacity (FVC) and forced expiratory capacity in 1 s (FEV1) were measured by spirometry (ParvoMedics Inc., Sandy, UT, USA). Maximal inspiratory pressure (MIP) was measured via the use of a digital pressure vacuum meter (Net Tech, Farmingdale, NY, USA).

Participants were ranked according to the results of the preliminary VO2peak data before being further divided into two groups, the control (non-ETM) group and the experimental (ETM-wearing) group, as mentioned above. The experimental group wore ETMs in all training sessions; the control group did not wear ETMs at any point. During the week before the training, both groups completed an identical training program. To familiarize the participants with the training protocol and equipment, all subjects completed two trial workouts. In the first session, the experimental group was required to wear an ETM for 10 min while sitting in a chair to familiarize them with how their breathing would be affected by the mask. Then, the experimental group was asked to ride on the mechanically braked cycle ergometers for 10 min at a pace of their own choosing. In the first practice session, the control group was also asked to ride the cycle ergometers for a 10 min period at a pace of their own choosing. In the second practice session, both groups were asked to complete five rounds of 30 s of cycling at peak PO, interspaced by 90 s of active recovery between intervals. ETMs were worn by the experimental group in these practice sessions, while the control group did not wear ETMs.

### 2.6. Training Program Protocol

The subjects completed an 8-week HIIT cycle ergometer program. The training sessions were held twice a week; each session lasted 35 min. The training was carried out using the same ergometers as in the pilot test. The workouts consisted of a 10 min warm-up; 20 min of HIIT; and a 5 min cool-down period (see Figure 1 below). The 20 min HIIT section featured 10 reps of 30 s at PPO (based on the final section of the VO2peak test) and then an active recovery period of 90 s, during which all participants maintained an output of 25 W. During each training session, subjects wore heart rate (HR) monitors, and HR readings were taken immediately after the high-intensity section of each interval. Meanwhile, the CR-10 modified Borg scale was used to gather ratings of perceived exertion immediately after the high-intensity sections of each interval. RPE was recorded on completion of the entire workout [15]. The intensity of the training over the 8-week course was set according to the participants’ RPE scores after interval ten during the workouts for each group: if the control group rated their two most recent consecutive intervals as ≤5 (hard), PO was increased by 0.5 kg (≈30 W) in the next training session. Similarly, if the control group and the experimental group rated their two most recent consecutive intervals as ≤7 (very hard) then the PO was increased by 0.5 kg (≈30 W) for the next training session. The pilot testing revealed that at identical workloads, the experimental group reported RPE as being two units higher than the control group. The experimental group wore ETMs during all training sessions. During weeks 1–2, the masks replicated exercising at 914 m above sea level; in weeks 3–4, the masks replicated exercising at 1829 m; during weeks 5–6, the masks simulated exercising at 2743 m; during weeks 7–8, the masks replicated exercising at 3658 m.

### 2.7. Statistical Analysis

The Statistical Package for Social Sciences version 26 for Windows (IBM, Armonk, NY, USA) was used. Since the measured variables did not follow a normal distribution after the Kolmogorov–Smirnov test was performed, nonparametric statistics were used. The Wilcoxon test was performed to compare the differences before and after training in each group. The Mann–Whitney U test was used to analyze the differences between variations in the dependent variables in the control and experimental groups and between initial variables in both groups. The linearity of pre-training variables was tested using scatterplots and multicollinearity using variance inflation factor (VIF) values, all of which were less than 5. The normality and homoscedasticity of the residuals were also verified. Stepwise multiple regression analysis was used to determine which variables significantly affected the variation in each variable. To avoid error type I, Bonferroni correction was applied to the *p*-value and set at *p* < 0.016.

## 3. Results

Of the 50 participants who began the study (control group: n = 25; experimental group: n = 25), only 44 successfully completed all 16 training sessions during the 8-week training period. If a participant missed a weekday session, a catch-up session was provided over the weekend. Three participants from each group were excluded from the analysis as they failed to complete the final training session and post-training VO2max testing because of irregular training patterns and/or injury.

At the outset of the study, all participants in the control and experimental groups were comparable in height, age, BMI, and weight (see Table 1 below).

Table 2 (below) illustrates that except for FEV₁, FEV₁/FVC, VT_1_, and MHR, significant differences were evident between the pre-test and post-test values of the control group for pulmonary function, hematological variables, and cardiorespiratory fitness biomarkers. The percentage ratios between the pre-test and post-test results of the control group for FVC, MIP, Hb, and Hct variables were −4.17%, 8.16%, −5.29%, and 3.45%, respectively. Cardiorespiratory fitness biomarkers including VO₂max, PPO, PO at VT_1_, VT_2_, and PO at VT_2_ changes were 2.10%, 9.32%, 7.42%, 2.10%, and 8.45%, respectively.

In the experimental group, compared to pre-test values, there were significant improvements in post-test values in all variables except FEV₁ ∕ FVC and HRM (Table 3). The percentage ratios between the pre-test and post-test results for FVC, FEV1, MIP, Hb, and Hct variables were 2.54%, 9.17%, 9.95%, −7.31%, and 2.70%, respectively. The percentage ratios of changes in VO₂max, PPO, VT1, PO at VT1, VT2, (PO at VT2), and HRpeak were also 11.32%, 16.10%, 12.32%, 17.80%, 12.30%, 17.65%, and 0.66%, respectively.

Table 4 and Table 5 (below) show the comparison of changes in pulmonary function, hematological data, and biomarkers of cardiorespiratory fitness after exercise between the control and experimental groups. Higher significant changes in FVC, FEV₁, VO₂max, VT_1_, PO to VT, and VT_2_ and PO to VT_2_ were observed in the experimental group. No significant difference was noted between the two groups in the remaining variables. Of note, significant differences were noted between the experimental and control groups in the initial levels of Hct and PO at VT_1_ and VT_2_ (Table 4).

## 4. Discussion

There were significant differences between the pulmonary function, hematological data, and biomarkers of cardiorespiratory fitness of the experimental group and the control group after the 8-week cycle ergometer HIIT program, in favor of the experimental group. Specifically, the experimental group achieved significant improvement in post-test values for all variables except FEV₁ ∕ FVC and HRM. The control group also experienced significant differences between the pre-test and post-test values in all parameters except for FEV₁, FEV₁/FVC, VT_1_, and MHR, with higher significant changes in FVC, FEV₁, VO₂max, VT1, PO to VT, VT2, and PO to VT2 in the experimental group compared to the control group.

The current study’s results concur with those of Kido et al. [16], who determined that the group using resistive breathing improved their VO2max scores by 18.5% and PPO by 11.1%; the control group (no resistive breathing) improved their VO2max by 11.7% and PPO by 11.5% [17]. However, the improvements in VO2peak and PPO did not differ significantly between the groups. The use of a resistive breathing mask during exercise also produced comparable improvements in VT1 (36%); no significant improvements in VT1 were observed in the exercise-only group [17]. Furthermore, HAT induces a state of hypoxia in the body due to lowered SpO2, known as high-altitude hypoxia or hypoxic hypoxia. Many researchers have investigated the physical benefits of alpine training during athletes’ general preparation for competitive endurance events [16,18]. Table 4 (above) shows that the use of ETMs contributed to significantly enhancing pulmonary function and hematological parameters in the experimental group as opposed to the control group.

The current study highlighted that the use of ETMs by the experimental group significantly improved the post-test pulmonary functions and hematological indicators compared to the control group. These results were dependent on the fairly lengthy 8-week training program, the participants’ fitness levels, and Saudi Arabia’s environment and climate. The present study’s results are at odds with Kido et al.’s [16] finding that no significant improvements were achieved in lung function between ETM-wearing and non-ETM-wearing groups measured via vital capacity, FVC, FEV1, and MIP [17]. Relatedly, Gething et al. [14] reported that athletes following a 6-week respiratory muscle training (RMT) course at 100% MIP had significantly decreased RPE. It was suggested that RMT may improve physiological capacity, which produces a reduction in the perceived cost of breathing. The results of the present study indicate that ETM wearing chiefly works to enhance respiratory muscle function via physiological adaptations to breathing that lead participants to perceive the task of breathing as easier when not wearing an ETM [4].

Our results are in line with several previous studies showing significant effects of altitude training on hematological indicators [19,20,21,22]. After three weeks of living and training at 1800 m altitude, a 3.0% increase in Hb mass was observed in one study [19]. Significant increases in hemoglobin mass associated with increases in reticulocytes and hematocrit compared to controls were also observed. Frese and Friedmann-Bette [20] also reported a significant increase in hemoglobin mass after three weeks of conventional high-altitude training at 1300 m and 1650 m alternated with training at sea level for three weeks. These authors also reported increased red blood cell volume, hemoglobin mass and serum EPO concentrations. Garvikan et al. [21], who examined Hb adaptation to erythropoietic evolution, reported a 1% increase per 100 h under natural or simulated conditions at moderate altitudes.

All participants increased their VO_2_peak due to the HIIT course (*p* = 0.001); both the experimental and control groups exhibited increased VO2peak on completion of the HIIT. It was hypothesized that the experimental group would achieve a greater improvement in VO2peak than the control group. The experimental group achieved a greater aggregated increase in VO2peak between pre- and post-testing than the control group; this difference between pre- and post-test VO2peak was statistically significant for both groups. This was also the case for FIVC and FVC values: significant improvements in FIVC and FVC were seen in both groups, with the experimental group achieving bigger improvements in FIVC and FVC. Besides, analyzing the pre- and post-test data at the participant level showed that predicted VO2peak and FVC significantly increased in all participants of both groups. Significant improvements were also seen individually for FIVC values. These data concur with previous studies on the benefits of adding HIIT sessions to existing exercise programs to enhance VO2peak in young, healthy, active adults, irrespective of their current level of fitness [6,15,22]. Biggs et al. [23] found results comparable to those of the present study by artificially inducing hypoxia by asking participants to breathe through gas masks (using hypoxic air). The results indicate that this had a beneficial effect on many physiological indicators, and primarily on the most crucial factor in endurance sports, VO2max [8].

## 5. Conclusions

The present study sought to apply a novel approach to experimentally investigate the efficacy of employing ETMs as part of an 8-week HIIT program. The results show that ETM use led to a range of significant improvements in hematological indicators and pulmonary function, VO2 peak, peak power output (PPO), cardiorespiratory fitness, lung function, and hematological variables in young, male, Saudi athletes. Training using ETMs at appropriate resistive breathing levels appears to be a safe and cost-effective way of enhancing performance in healthy athletes without costly altitude training. Further investigations are encouraged to interpret the physiological changes produced by ETM-assisted HIIT programs.

## Figures and Tables

**Figure 1 ijerph-20-03533-f001:**
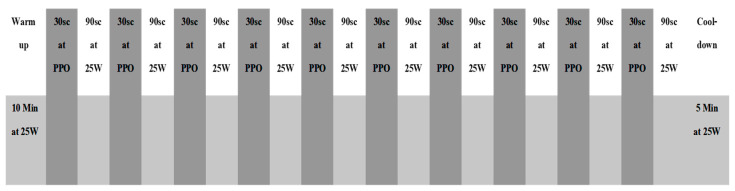
Training program protocol, including warm-up and cool-down.

**Table 1 ijerph-20-03533-t001:** Physical characteristics of both groups (experimental group: n = 22; control group: n = 22) presented as mean ± SD.

Variables	With-ETM	Non-ETM
Age (years)	20.56 ± 1.01	21.81 ± 0.21
Height (cm)	180.89 ± 2.79	179.82 ± 2.39
Weight (Kg)	79.21 ± 3.19	78.13 ± 2.41
Body Mass Index	24.27 ± 1.28	24.17 ± 1.03

**Table 2 ijerph-20-03533-t002:** Pulmonary function, hematological variables, and cardiorespiratory fitness biomarkers: a comparison of the pre- and post-test changes in the control group.

Variables	N	Pre-Training	Post-Training	Changes	Z	*p*
FVC (L)	22	5.28 ± 0.53	5.06 ± 0.48	−0.22 ± 0.14	−4.051	0.001
FEV₁ (L)	22	4.35 ± 0.13	4.38 ± 0.01	0.03 ± 0.13	−0.991	NS
FEV₁ (L)/FVC (L)	22	81.91 ± 4.55	82.81 ± 3.05	0.90 ± 3.28	−2.171	NS
MIP (cmH₂o)	22	80.85 ± 14.96	87.45 ± 9.39	6.60 ± 11.10	−3.589	0.001
Hb (g/dL)	22	13.26 ± 0.33	14.00 ± 0.48	0.75 ± 0.35	4.051	0.001
Hct (%)	22	42.00 ± 0.97	43.44 ± 0.18	1.44 ± 0.92	−4.113	0.001
VO₂max (mL/kg/min)	22	46.58 ± 2.14	47.58 ± 2.14	1.00 ± 3.09	−3.428	0.001
PPO (watts)	22	276.68 ± 27.66	302.50 ± 23.89	25.82 ± 25.75	−3.702	0.001
VT1 (mL/kg/min)	22	29.18 ± 3.41	29.31 ± 2.55	0.13 ± 4.48	−0.087	NS
PO at VT (watts	22	158.09 ± 11.11	169.82 ± 10.49	11.73 ± 5.61	−4.017	0.001
VT2 (mL∙kg^−1^ min^−1^)	22	38.70 ± 4.24	39.48 ± 3.27	0.78 ± 1.24	−4.107	0.001
PO at VT2 (watts)	22	254.82 ± 17.26	266.59 ± 19.57	11.77 ± 26.14	−3.320	0.001
MHR (bpm)	22	183.41 ± 8.91	184.57 ± 8.68	1.16 ± 2.37	−2.070	NS

Note: Data represented as mean ± SD. NS, not significant. The difference was significant at *p* < 0.016 using the Wilcoxon test. Forced vital capacity, FVC; forced expiratory volume, FEV; hematocrit, Hct; hemoglobin, Hb; maximum oxygen consumption rate VO2 max; peak power output PPO; ventilatory threshold VT; maximal heart rate (MHR).

**Table 3 ijerph-20-03533-t003:** Pulmonary function, hematological variables, and cardiorespiratory fitness biomarkers: a comparison of the pre- and post-test changes in the experimental group (n = 22).

Variables	N	Pre-Training	Post-Training	Changes	Z	*p*
FVC (L)	22	5.11 ± 0.44	5.24 ± 0.34	0.12 ± 0.11	−4.156	0.001
FEV₁ (L)	22	4.36 ± 0.22	4.76 ± 0.14	0.40 ± 0.22	−4.076	0.001
FEV₁ (L)/FVC (L)	22	82.14 ± 5.68	83.17 ± 4.42	1.03 ± 3.45	−2.363	NS
MIP (cmH₂o)	22	80.23 ± 14.59	88.21 ± 8.76	7.98 ± 10.75	−4.130	0.001
Hb (g/dL)	22	12.80 ± 0.41	13.80 ± 0.68	1.00 ± 0.47	4.047	0.001
Hct (%)	22	41.27 ± 1.56	42.37 ± 1.48	1.10 ± 0.95	−3.948	0.001
VO₂max (mL/kg/min)	22	46.80 ± 3.19	52.09 ± 3.17	5.29 ± 1.91	−4.108	0.001
PPO (watts)	22	269.37 ± 29.98	312.73 ± 25.74	43.36 ± 29.27	−4.075	0.001
VT1 (mL/kg/min)	22	29.47 ± 4.28	33.10 ± 3.13	3.64 ± 3.62	−3.820	0.001
PO at VT (watts)	22	162.23 ± 14.18	191.50 ± 14.76	29.27 ± 10.12	−4.113	0.001
VT2 (mL∙kg^−1^ min^−1^)	22	39.00 ± 5.11	43.80 ± 3.33	4.80 ± 4.05	−4.108	0.001
PO at VT2 (watts)	22	240.13 ± 22.34	282.52 ± 21.93	42.40 ± 19.78	−4.107	0.001
MHR (bpm)	22	183.59 ± 9.33	184.73 ± 8.07	1.14 ± 4.50	−0.647	NS

Note: Data represented as mean ± SD. NS, not significant. The difference was significant at *p* < 0.016 using the Wilcoxon test. forced vital capacity, FVC; forced expiratory volume, FEV; hematocrit, Hct; hemoglobin, Hb; maximum oxygen consumption rate VO2 max; peak power output PPO; ventilatory threshold VT; maximal heart rate (MHR).

**Table 4 ijerph-20-03533-t004:** Comparison of baseline values of pulmonary and hematological functions and biomarkers of cardiorespiratory fitness between the experimental and the control groups.

Variables	N	Control Group	Experimental Group	Z	*p*
FVC (L)	22	5.28 ± 0.53	5.11 ± 0.44	−2.480	NS
FEV₁ (L)	22	4.35 ± 0.13	4.36 ± 0.22	−0.459	NS
FEV₁ (L)/FVC (L)	22	81.91 ± 4.55	82.14 ± 5.68	−0.400	NS
MIP (cmH₂o)	22	80.85 ± 14.96	80.23 ± 14.59	−0.413	NS
Hb (g/dL)	22	14.00 ± 0.48	13.80 ± 0.68	−1.016	NS
Hct (%)	22	42.00 ± 0.97	41.27 ± 1.56	−2.911	0.004
VO₂max (mL/kg/min)	22	46.58 ± 2.14	46.80 ± 3.19	−2.038	NS
PPO (watts)	22	276.68 ± 27.66	269.37 ± 29.98	−1.257	NS
VT_1_ (mL/kg/min)	22	29.18 ± 3.41	29.47 ± 4.28	−0.611	NS
PO at VT_1_ (watts)	22	158.09 ± 11.11	162.23 ± 14.18	−2.753	0.006
VT_2_ (mL∙kg^−1^ min^−1^)	22	38.70 ± 4.24	39.00 ± 5.11	−2.335	NS
PO at VT_2_ (watts)	22	254.82 ± 17.26	240.13 ± 22.34	−3.748	0.001
MHR (bpm)	22	183.41 ± 8.91	183.59 ± 9.33	−1.256	NS

Note: Data represented as mean ± SD. NS, not significant. The difference was significant at *p* < 0.016 using the Mann–Whitney U test. Forced vital capacity, FVC; forced expiratory volume, FEV; hematocrit, Hct; hemoglobin, Hb; maximum oxygen consumption rate VO2 max; peak power output PPO; ventilatory threshold VT; maximal heart rate (MHR).

**Table 5 ijerph-20-03533-t005:** Comparison of changes in pulmonary function, hematological data, and biomarkers of cardiorespiratory fitness after eight weeks of high-intensity interval training with (experimental group) or without (control group) use of an elevation mask.

Variables	Groups	N	Changes	Z	*p*
FVC (L)	Control Group	22	−0.22 ± 0.14	−5.74	0.001
	Experimental Group	22	0.12 ± 0.11		
FEV₁ (L)	Control Group	22	0.03 ± 0.13	−4.990	0.001
	Experimental Group	22	0.40 ± 0.22		
FEV₁ (L)/FVC (L)	Control Group	22	0.90 ± 3.28	−1.222	NS
	Experimental Group	22	1.03 ± 3.45		
MIP (cmH₂o)	Control Group	22	6.60 ± 11.10	−1.122	NS
	Experimental Group	22	7.98 ± 10.75		
Hb (g/dL)	Control Group	22	0.75 ± 0.35	−2.346	NS
	Experimental Group	22	1.00 ± 0.47		
Hct (%)	Control Group	22	1.44 ± 0.92	−1.755	NS
	Experimental Group	22	1.10 ± 0.95		
VO₂max (mL/kg/min)	Control Group	22	1.00 ± 3.09	−5.167	0.001
	Experimental Group	22	5.29 ± 1.91		
PPO (watts)	Control Group	22	25.82 ± 25.75	−2.313	NS
	Experimental Group	22	43.36 ± 29.27		
VT1 (mL/kg/min)	Control Group	22	0.13 ± 4.48	−3.336	0.001
	Experimental Group	22	3.64 ± 3.62		
PO at VT (watts)	Control Group	22	11.73 ± 5.61	−5.359	0.001
	Experimental Group	22	29.27 ± 10.12		
VT2 (mL∙kg^−1^ min^−1^)	Control Group	22	0.78 ± 1.24	−5.153	0.001
	Experimental Group	22	4.80 ± 4.05		
PO at VT2 (watts)	Control Group	22	11.77 ± 26.14	−4.508	0.001
	Experimental Group	22	42.40 ± 19.78		
MHR (bpm)	Control Group	22	1.16 ± 2.37	−1.299	NS
	Experimental Group	22	1.14 ± 4.50		

Note: Data represented as mean ± SD. NS, not significant. The difference was significant at *p* < 0.016 using the Mann–Whitney U test. Forced vital capacity, FVC; forced expiratory volume, FEV; hematocrit, Hct; hemoglobin, Hb; maximum oxygen consumption rate VO2 max; peak power output PPO; ventilatory threshold VT; maximal heart rate (MHR).

## Data Availability

Data are available upon request to the main author.

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
