# Peer review of "Eight Weeks of High-Intensity Interval Training Using Elevation Mask May Improve Cardiorespiratory Fitness, Pulmonary Functions, and Hematological Variables in University Athletes"

_ijerph, 2023, doi:10.3390/ijerph20043533_

Round 1

Reviewer 1 Report

1. I suggest some description above hemotological change of high attitude training and ETM training.

2. Line 87-91: Data is not propriate shown in the Method section. You just need to say " The participants were divided into two comparable groups by their age and sports. ". And data should be moved to the beginning of the Result section.

3.Line 94: should be “ … based on work by Lundby et al. (2016) “.

4. Line 108: should be “ Kg/m2

5. Line 165:  Is “ control and experimental group “, or “ experimental group ” ?

6. Line 179: should be “ paired t-test “

7. Line 197-199: Writing is not compatible to Table 2.

8. Why Hb decreased after training? Hb increases after aerobic training and high attitude training. Is similar change revealed in previous study? Please explain it in the Discussion section.

9. Table 3: What is RCT ? General speaking, a table or a figure is self-talking, and I suggest abbreviation list in the legend of the table.

10. Table 4 and 5: I suggest to merge tow tables into one table. Besides, “ changes “ in the table title should be “ values “, and “ Change “ in the table should be “ Difference “.

11. In table 5, did you lose “ * “ over VO2max and PPO in Non-ETM column ?

12. Line 254-256: Writing is not compatible to Table 3.

13. Line 263-265: Need further statistical test to say this sentence. You may express this significance in “ Difference “ column of Table 4 and 5, if you prove it.

14. Line 291-295: I think this section is a duplicate of result. I suggest to delete it.

Author Response

Responses to reviewers’ comments

Reviewer 1

Comment: I suggest some description of the hematological change of high altitude training and ETM training.

Response: Hematological changes after high altitude training and ETM training were added.

Comment: Line 87-91: Data is not appropriately shown in the Method section. You just need to say " The participants were divided into two comparable groups by their age and sports. ". And data should be moved to the beginning of the Result section.

Response: Modified as follows: the participants were divided into two comparable groups by their age and sports." And we moved “Group 1 (experimental group) ETM-wearers (N = 20; runners: n = 7; swimmers: n = 6; cyclists: n = 7; aged 21.24±0.14 years old mean±SD); Group 2 (control group) non-ETM wearers (N = 20; runners: n = 7; swimmers:  n = 6; cyclists: n = 7; aged 21.35±0.19 years old mean±SD).

Comment: Line 94: should be “… based on work by Lundby et al. (2016) “.

Response: Changed.

Comment: Line 108: should be “Kg/m2 “

Response: Changed.

Comment: Line 165:  Is “control and experimental group “, or “experimental group”?

Response: Both groups.

Comment: Line 179: should be “paired t-test “

Response: The statistical tests have been revised and this sentence has been removed.

Comment: Line 197-199: Writing is not compatible with Table 2.

Response: This comment has been revised.

Comment: Why Hb decreased after training? Hb increases after aerobic training and altitude training. Is a similar change revealed in the previous study? Please explain it in the Discussion section.

Response: We apologize; this is a typo. A separate paragraph has been added to the Discussion section, reporting the effects of altitude training on Hb. Thank you for your comment.

Comment: Table 3: What is RCT ? General speaking, a table or a figure is self-talking, and I suggest abbreviation list in the legend of the table.

Response: all abbreviations were added.

Comment: Table 4 and 5: I suggest to merge tow tables into one table. Besides, “changes “ in the table title should be “ values “, and “ Change “ in the table should be “ Difference “.

Response: Many changes have been made to the tables.

Comment: In table 5, did you lose “ * “ over VO2max and PPO in Non-ETM column ?

Response: The p values have been added to the tables.

Comment: Line 254-256: Writing is not compatible to Table 3.

Response: corrected.

Comment: Line 263-265: Need further statistical test to say this sentence. You may express this significance in “ Difference “ column of Table 4 and 5, if you prove it.

Response: All statistics were reviewed and corrected.

Comment: Line 291-295: I think this section is a duplicate of result. I suggest to delete it.

Response: Deleted.

Reviewer 2 Report

Please refer to the documents attached.

Author Response

Responses to reviewers’ comments

Reviewer 2

Comment: Line 56-58: citation needed. Which studies?

Response: The statement is revised “Previous studies investigated the impact of wearing ETMs during training programs in amateur runners and cyclists participating in HIIT and Reserve Officers' Training Corps (ROTC) cadets undergoing physical training (PT) [7–12].”

Comment: Line 63-64: citation needed. Which studies?

Response: The statement is revised “Previous studies investigated the impact of wearing ETMs during training programs in amateur runners and cyclists participating in HIIT and Reserve Officers' Training Corps (ROTC) cadets undergoing physical training (PT) [7–12].” HITT is HIIT and it is corrected.

Comment: Line 76-84: citation needed. Which studies?

Response: The statement is revised with more explanation “Enhancing the efficacy of such training requires constant research to develop novel evidence-based methods including using diagnostic approaches to gather data on functional efficiency indicators. This is particularly relevant to elite athletes: Any slight advantage over their competitors conferred by novel training methods can be critical for success. In light of the above, the present study aimed to investigate the impact of eight weeks of HIIT using ETM on physiological and hematological variables in university athletes. This experimental study will be the first phase of the project that will compare the outcomes of the present study with the outcomes of the second phase that apply simulated altitude hypoxic using an environmental chamber in the same participants.”

Comment: Line 106-107: a weight scale?

Response: Yes, and it has been clarified in the statement “and weight (to the nearest 0.1 kg) were measured using Seca weight scale (Seca-869, Germany).”
